# Enhanced Antitumor Efficacy of Cytarabine and Idarubicin in Acute Myeloid Leukemia Using Liposomal Formulation: In Vitro and In Vivo Studies

**DOI:** 10.3390/pharmaceutics16091220

**Published:** 2024-09-19

**Authors:** Chunxia Zhu, Yang Liu, Xiaojun Ji, Yaxuan Si, Xianhao Tao, Xiaohua Zhang, Lifang Yin

**Affiliations:** 1Department of Pharmaceutics, China Pharmaceutical University, Nanjing 210009, China; xznn_ly@126.com; 2Department of Pharmaceutics, Nanjing Chia-Tai Tianqing Pharmaceutical Co., Ltd., Nanjing 210046, China; rmw5157@dingtalk.com (Y.L.); xzmcjxj@dingtalk.com (X.J.); siyaxuan@dingtalk.com (Y.S.); txh_086@163.com (X.T.); yyzxh81@163.com (X.Z.); 3Key Laboratory of Drug Quality Control and Pharmacovigilance, China Pharmaceutical University, Nanjing 210009, China; 4State Key Laboratory of Natural Medicine, China Pharmaceutical University, Nanjing 210009, China; 5NMPA Key Laboratory for Research and Evaluation of Pharmaceutical Preparations and Excipients, China Pharmaceutical University, Nanjing 210009, China

**Keywords:** dual-loaded liposome, cytarabine, idarubicin, acute myeloid leukemia

## Abstract

**Background:** Acute myeloid leukemia (AML) is the most common type of acute leukemia among adults with the recommend therapy of combination of cytarabine and idarubicin in the induction phase. The uncoordinated pharmacokinetics prevent adequate control of drug ratio following systemic administration. Therefore, the dual-loaded liposomes containing cytarabine and idarubicin for synergistic effects were proposed and investigated. **Methods:** The molar ratio of cytarabine and idarubicin for synergistic effects was investigated. The dual-loaded liposomes were prepared and characterized by particle size, zeta potential, encapsulation efficiency, cryo-Transmission electron microscopy (cryo-TEM), and in vitro stability. The in vitro cytotoxicity and cell uptake of liposomes were determined within CCRF-CEM cells. The PK experiments was carried out in male SD rats. The in vivo antitumor effect was carried out within CD-1 nude female mice. The antitumor mechanism of liposomes was investigated. **Results:** The synergistic molar ratios were found to be in the range of 20:1~40:1. The size distribution of the dual-loaded liposomes was approximately 100 nm with PDI ≤ 0.1, a zeta potential of approximately −30 mV, an entrapment efficiency of cytarabine and idarubicin of >95% with spherical structure and uniform distribution, and in vitro stability for 21 d. The drugs in the liposomes can be quickly uptaken by the leukemia cells. The PK experiments showed that the molar ratio of cytarabine to idarubicin in plasma was maintained at 30:1 within 4 h. The efficacy of liposomes was significantly enhanced. Conclusions: The dual-loaded liposomes containing cytarabine and idarubicin showed enhanced antitumor efficacy.

## 1. Introduction

Acute myeloid leukemia (AML), as an aggressive blood and bone marrow cancer, is characterized by the uncontrolled proliferation of bone marrow stem cells [1]. It is the most common type of acute leukemia among adults, with over 19,900 new cases diagnosed and 11,100 deaths registered in the United States during 2020 [2,3]. AML treatment is extremely expensive and challenging, with a five-year survival rate of approximately 20% overall, which decreases sharply in older patients [4], causing great hardship for patients and their families.

Combination therapies have been proven to be more effective in treating cancer patients compared to monotherapies. This can be attributed to the theoretical rationale including enhancing therapeutic effects, reducing toxicity, minimizing adverse reactions, and delaying the development of drug resistance [5]. The action patterns of combined drugs may be either synergistic, additive, or antagonistic depending on the ratio of the agents being combined [6]. The combination index (CI) is used directly to reflect the extent of synergy between drugs [7]. CI values of <1 are indicative of a synergistic effect [8].

Combination therapy remains the standard of care for intensive AML induction therapy, consisting of a multiple-day infusion of cytarabine in combination with anthracycline [9]. Specifically, the “7 + 3” regimen of 7 days of continuous intravenous (IV) cytarabine along with the first 3 concurrent days of intravenous daunorubicin/idarubicin has been broadly accepted as a standard of care treatment since 1973 [10]. In light of the substantial body of evidence-based data on efficacy and safety, this combination therapy remains the recommended first-line treatment for AML induction in the latest edition of the NCCN guideline (Version 3, 2024). Due to variations in drug properties, the uncoordinated pharmacokinetics of cytarabine and daunorubicin/idarubicin cocktails prevent adequate control of the drug ratio following systemic administration [11], which may limit therapeutic outcomes. As research has proposed, achievement and maintenance of cytarabine/daunorubicin in the 5:1 molar ratio maximizes efficacy, but the synergistic ratio must be delivered to the malignant cells effectively [6,9]. However, the traditional combination of the “7 + 3” regimen may be insufficient to achieve and maintain a particularly efficacious molar ratio. For example, the plasma concentration of cytarabine stays relatively constant during 7 days of continuous infusion, while the plasma concentration of daunorubicin/idarubicin changes after IV infusion on days 1, 2, and 3, resulting in a constantly changing molar ratio within the plasma as well as malignant cells [9]. The traditional combination of the “7 + 3” regimen remains dismal in older (>70 years) patients, as the low rates of complete remission in older patients diagnosed with increased prevalence of AML-specific genomic and cytogenetic risk factors are associated with a poor prognosis [9,12]. Therefore, an ideal drug delivery system is needed to precisely control the in vivo delivery and release processes of the different antitumor drugs, making synergistic pharmaceuticals.

Liposomes have been suggested as a suitable candidate for the delivery of drug combinations while maintaining their synergetic ratios, coordinating pharmacokinetic differences, and prolonging in vivo retention time [13]. The unique bilayer structure of liposomes enables them to simultaneously contain hydrophilic and lipophilic drugs. Vyxeos, the only dual-drug-loaded liposome, encapsulates cytarabine and daunorubicin at an optimal synergistic molar ratio of 5:1 for treating AML, approved by the FDA in 2017. The dual-loaded liposome is superior to the conventional “7 + 3” regimen in the treatment of newly diagnosed therapy-related AML or AML with myelodysplasia-related changes in adults and pediatric patients 1 year and older [14,15,16,17]. The development of Vyxeos demonstrates the benefits of rational combination therapy design.

A combination regimen of cytarabine and idarubicin is also one of the preferred induction therapeutics for AML. Compared with daunorubicin, idarubicin has higher activity, less cardiotoxicity, and less sensitivity to the activities of P-gp and multidrug-resistant proteins [18], which indicates the potential for improved effectiveness when idarubicin is encapsulated in liposomes. In the studies presented here, we investigated the dual-loaded liposomes of cytarabine and idarubicin for the treatment of AML.

However, earlier data suggested the increased leakage of idarubicin from cholesterol-containing liposomes than from cholesterol-free liposomes [19,20,21], attributing to the domain-separating boundaries [22]. Recent data show that the formulation of stably insoluble drug precipitates inside liposomes can lead to increased drug retention equally, even if cholesterol-containing liposomes are used [23,24]. What is more, cholesterol plays a strategic role in liposomal composition [25], as demonstrated by reducing bilayer permeability, along with improving vesicle stability both in vitro and in vivo [26] and changing the fluidity of the lipid bilayer. Therefore, a low-cholesterol formulation (10% molar/molar) was used in our studies. Subsequently, to improve stability and drug retention, the gel-phase liposomes were made with saturated acyl side chain phosphatidyl lipids [27], such as DSPC, a long-chain phospholipid with a high phase transition temperature, to increase stability at body temperature [28]. DSPG, a negatively charged phosphatidylglycerol, was used to increase the circulation longevity of the liposomes, clarified to minimize aggregation by Coulombic repulsive force [29,30]. In our previous work, the lipid membrane of the dual-loaded liposomes composed of DSPC/DSPG/Chol at a molar ratio of 7/2/1 could effectively encapsulate both cytarabine and idarubicin. As DSPG was not soluble in ethanol, the ethanol injection method was not employed. Consequently, the dual-loaded liposomes of cytarabine and idarubicin were prepared by sequential and simultaneous encapsulation methods. Firstly, the cytarabine was loaded using a passive-loading method with thin film hydration. Subsequently, idarubicin was encapsulated using the active-loading method [6,28].

In our research, we identified the synergistic molar ratios in various cell lines and created a dual-loaded liposome formulation to enhance the retention and release of idarubicin. The dual-loaded liposomes of cytarabine and idarubicin were characterized for in vitro and in vivo aspects.

## 2. Materials and Methods

### 2.1. Materials

1,2-distearoyl-sn-glycero-3-phosphocholine (DSPC) and 1,2-distearoyl-sn-glycero-3-phosphoglycerol (DSPG) were purchased from Lipoid GmbH (Ludwigshafen, Germany). Cholesterol (Chol) was purchased from AVT (Shanghai) Pharmaceutical Tech Co., Ltd (Shanghai, China). Cytarabine was purchased from Fujian South Pharmaceutical Co., Ltd. (Fujian, China). Idarubicin hydrochloride was obtained from Nanjing Chia-Tai Tianqing Pharmaceutical Company (Nanjing, China). Copper gluconate, iron gluconate dihydrate, ammonium sulfate, triethanolamine, and 4-2-hydroxyethyl-1-piperazineethanesulfonic acid (HEPES) were purchased from Aladdin (Shanghai, China). Chloroform, methanol, EDTA-Na, citric acid monohydrate, triethanolamine, and trisodium citrate dihydrate were purchased from Nanjing Chemical Reagent Co., Ltd. (Nanjing, China). Sucrose was purchased from Hunan Jiudian Pharmaceutical Co., Ltd. (Hunan, China). Roswell Park Memorial Institute 1640 (RPMI-1640), Iscove’s Modified Dulbecco’s Dulbecco’s Medium (IMDM), McCoy’s 5A Medium, Fetal Bovine Serum (FBS), and Hank’s Balanced Salt Solution (HBSS) were purchased from Gibco (Logan City, UT, USA); Dimethyl sulfoxide (DMSO) and 2-mercaptoethanol were purchased from Sigma-Aldrich (Merck, Darmstadt, German); CellTiter-Glo luminescent cell viability assay was purchased from Promega (Madison, WI, USA); a cell cycle and apoptosis detection kit and sucrose were purchased from Beyotime (Shanghai, China); a FastPure cell/tissue total RNA isolation kit and Taq Pro Universal SYBR qPCR master Mix were purchased from Vazyme (Nanjing, China); First-Strand cDNA Synthesis Su-perMix was purchased from Transgene (Beijing, China); Phosphate Buffered Saline (PBS) was purchased from YEASEN (Shanghai, China). Lipo-1 (10:1), Lipo-2 (10:1), Lipo-3 (10:1), Lipo-4 (20:1), Lipo-5 (30:1), and Lipo-6 (40:1) were made by Nanjing Chia-Tai Tianqing Pharmaceutical Company (Nanjing, China).

### 2.2. Cell Cultures

During the experiment, a total of 13 cell lines were used (Table 1), including human acute T lymphoblastic leukemia cells Molt-4 and CCRF-CEM, human myeloid leukemia cells MV-4-11, human acute myeloid leukemia cells KG-1, OCI-AML-3, Kasumi-1, and MOLO-13, human acute promyelocytic leukemia cells HL-60, human colorectal carcinoma cells HCT116, human pancreatic carcinoma cells KP-4, human ovarian cancer cells TOV-21G and ES-2, and murine myeloid leukemia cells WEHI-3B. All cell lines except ES-2 were obtained from Cobioer Biosciences Co., Ltd. (Nanjing, China) and ES-2 was obtained from the National Collection of Authenticated Cell Cultures (Shanghai, China). The components of the cell culture medium are shown in Table 1; cells were maintained at 37 °C in a humidified atmosphere containing 5% CO_2_.

### 2.3. Animals

Female CD-1 mice and male Sprague-Dawley rats were used for pharmacokinetic analysis. Mice were purchased from Charles River (Jiaxing, China) aged between 3 and 5 weeks old, and rats from Jiangsu Value Pharmaceutical Services Co. LTD (Nanjing, China). aged between 7 and 11 weeks old, and weighed 200 g to 300 g. Female CD-1 nude mice, purchased from Charles River (Beijing, China) at ages of 5 to 7 weeks old, were used for in vivo antitumor evaluation.

The animals were housed in a barrier environment with free access to food and water. The tests were started after 1 week of acclimatization to the environment. Animal welfare and experimental procedures were carried out following the Guide for the Care and Use of Laboratory Animals (National Institutes of Health, Bethesda, MD, USA). The animal experimental protocols of pharmacokinetics and antitumor efficacy were approved by the Animal Ethics Committee of Value Pharmaceutical Services Co., Ltd. (Nanjing, China, Ethics number: AP-202107) and 3D BioOptima New Drug Development Co., Ltd. (Suzhou, China, Ethics number: 3D_ACUC2022012).

### 2.4. In Vitro Anti-Tumor Effect of Drug Combinations

The in vitro anti-tumor effect of cytarabine and idarubicin was determined by using a CellTiter-Glo luminescent cell viability assay. The leukemia cells were used primarily to evaluate the synergistic effect of the combination of cytarabine and idarubicin, and four additional solid tumor cells were selected to see if the combination effect was cell-type specific. The cells were seeded in 96-well plates with an appropriate complete cell culture medium; the cell number seeded in each well is shown in Table 1. Cells were treated with cytarabine and idarubicin, respectively, as well as with a combination of these two compounds. The concentration started at 30 μM, with a dilution ratio of 3 and a total of 10 concentrations, and each concentration had two wells. As for the combination group, the molar ratio of cytarabine to idarubicin was 5:1, 10:1, 20:1, 30:1, 40:1, and 50:1. Cytarabine and idarubicin were dissolved in DMSO, and DMSO was used as the negative control, the maximal concentration of which was 0.3% (*v*/*v*). The compounds were added using a Tecan D300e compound titrator. The cells were cultured for 72 h, and the CTG solution was added directly to the wells. Then, the plates were vibrated for 5 min and incubated for 20 min. The luminescence value was determined using a Decan SPARK multifunctional enzyme marking instrument.

### 2.5. Preparation of Liposomes

Liposomes consisting of DSPC/DSPG/Chol (7:2:1 mol/mol) were prepared using the thin-film hydration method followed by extrusion [31,32]. Briefly, 1500 mg of lipids used for each sample were dissolved in chloroform/methanol/water (95:4:1 *v*/*v*, 50 mg/mL) in a 100-mL round-bottom flask. The resultant solution was subjected to rotary vacuum evaporation at 60 °C with IKA RV10 (Staufenim, Germany) to form a thin film on the inner wall of the flask. The residual solvent was removed under a high vacuum for at least 1 h. Multilamellar vesicles (MLVs) were formed by hydrating the lipid film with 30 mL of appropriate solutions of cytarabine/TEA/Glu-Cu^2+^ (205 mM/220 mM/100 mM, pH 7.4), cytarabine/TEA/Glu-Fe^2+^ (205 mM/220 mM/100 mM, pH 7.4), or ammonium sulfate (300 mM, pH 5.5). Large unilamellar vesicles (LUVs) were prepared by extrusion at 55 °C through Nucleopore polycarbonate filters (Whatman, Inc., Clifton, NJ, USA) with pore sizes of 200 nm 4 times, 80 nm 4 times, and 50 nm 3 times on an extruder (Lavifluid, Shanghai, China), respectively. The mean diameter of the vesicles was measured with a Nanobrook Omni particle size analyzer (Brookhaven, New York, NY, USA).

### 2.6. Preparation of the Ion Gradient for Idarubicin Encapsulation

The ion gradient [33] was subsequently generated by exchanging the extravesicular liposomal solution using a tangential flow filtration system (Repligen, Boston, MA, USA) with 2 mM EDTA at a 10-fold volume of the liposomes and then ultrafiltrated with a solution containing 10% sucrose and 20 mM HEPES at pH 7.4. The concentration of phospholipids was quantified using Agilent HPLC-ELSD (Santa Clara, CA, USA). The encapsulation efficiency of cytarabine was determined using a Sephadex G-50 of Cytiva (Marlborough, MA, USA) column and was quantified using HPLC-UV (Agilent, CA, USA).

Idarubicin was loaded into preformed liposomes as follows. Idarubicin hydrochloride solution in water (5 mg/mL) was added to the LUV suspensions to achieve molar ratios of 10:1, 20:1, 30:1, and 40:1 of cytarabine/idarubicin. The loading process was carried out at 50 °C for 30 min. The mean diameter of the vesicles was measured with a Nanobrook Omni particle size analyzer (Brookhaven, New York, NY, USA). The encapsulation efficiency of cytarabine and idarubicin was determined using a Sephadex G-50 (Cytiva, USA) column and was quantified using HPLC-UV (Agilent, CA, USA).

### 2.7. Characterization of Liposomes

#### 2.7.1. Size Distribution and Zeta Potential

The particle size and zeta potential of the liposome formulations were measured by dynamic light scattering analysis using a Nanobrook Omni particle size analyzer (Brookhaven, New York, NY, USA). For the measurement of particle size, the liposome suspensions were diluted by a neutral phosphate sucrose solution into 1 mg/mL of lipids. For the zeta potential measurements, the sample cell was filled with sample solution to ensure the electrode was covered. The sample temperature was equilibrated to 25 °C for both measurements. The particle size distribution plots in the intensity-weighted mode were obtained using BIC Particle Solutions software (version 3.6.0.7122) supplied with the instrument.

#### 2.7.2. Determination of Encapsulation Efficiency (EE)

The encapsulation efficiency (EE) of cytarabine and idarubicin was determined using a Sephadex G-50 mini-column (1.5 × 30 cm) [34]. Non-encapsulated drugs were removed from the liposomes with citrate buffer containing 64.5 mM citric acid monohydrate and 150 mM trisodium citrate dihydrate at pH 7.4. Then, 100 μL liposome samples were placed on the column, and the free cytarabine and idarubicin were collected into a 10 mL flask, and the encapsulated cytarabine and idarubicin were collected into a 25 mL flask, followed by solubilization with 2% Triton-X 100 (containing 10 μL/mL ammon solution). The concentration of cytarabine and idarubicin was determined at 254 nm using HPLC-UV, and all samples were analyzed in triplicate.

The EE(%) was calculated as follows:(1)$${\mathrm{EE}}_{\mathrm{drug}}(\%)=\frac{{Drug}_{encapsulated}}{{Drug}_{free}+{Drug}_{encapsulated}}\times 100\%$$
where *Drug_encapsulated_* and *Drug_free_* are the encapsulated weight and the free weight of cytarabine or idarubicin, respectively.

#### 2.7.3. Cryo-Transmission Electron Microscopy (Cryo-TEM)

Liposomal samples were vitrified on copper grids coated with holey carbon 200 mesh (Quantifoil, Jena, Germany). About a 3~5 μL drop was applied to a grid and blotted with a filter paper to form a thin liquid film of solution [35]. Blotted samples were immediately plunged into liquid ethane just above its freezing point (−183 °C) using Vitrobot. The vitrified samples were stored under liquid nitrogen before being transferred to a Thermo FEI Talos F200C TEM (Waltham, MA, USA) using a cryo-holder for imaging at about −183 °C. The microscope was operated at 200 KV in a low electron dose mode (to reduce radiation damage) and with a few micrometers under focus to increase phase contrast. The images were recorded with a fast 4 K × 4 K Thermo Scientific Ceta 16M camera supported by Thermos Scientific Maps Software (Version 3.0).

#### 2.7.4. Evaluation of Stability of Liposomes In Vitro

The liposome formulations were prepared and stored in a refrigerator at 4 °C. Samples were taken at 0, 7, 14, and 21 days. The appearance of the preparation was observed, and the particle size distribution, the zeta potential, and the encapsulation efficiency (EE) were measured to evaluate the stability of liposomes.

### 2.8. Evaluation of Cell Uptake and Cytotoxicity of Liposomes In Vitro

The in vitro cytotoxicity of liposomes was determined using a CellTiter-Glo luminescent cell viability assay. CCRF-CEM cells were seeded in 96-well plates (8000 cells per well) with an appropriate complete cell culture medium. Cells were treated with Lipo-5 (30:1) and the combination of two free drugs (cytarabine and idarubicin). Lipo-5 was diluted with 10% sucrose, which was used as the negative control. The two free drugs were dissolved with DMSO, and DMSO was used as the negative control with a maximal concentration of 0.4% (*v*/*v*). Cells were cultured for 24 h, and the CTG solution was added directly to the wells, then the plates were vibrated for 5 min and incubated for 20 min. The luminescence value was determined by using a Decan SPARK multifunctional enzyme marking instrument.

In vitro cell uptake was evaluated by confocal microscopy. DiD dye was diluted to 5 μM with RPMI1640 medium, and cells were stained for 15 min. After the staining was completed, CCRF-CEM cells were washed with medium and total 2E5 cells were incubated with Lipo-1 for 0.5 h, 2 h, 4 h, 8 h, and 16 h with an appropriate complete cell culture medium, in which the concentration of Lipo-1 was 500 μM. After incubation, cells were centrifuged to remove the supernatant, and the cells were stained by Hoechst dye at 37 °C for 0.5 h. After that, the supernatant was removed by centrifugation, and cells were washed 3 times with cold HBSS and visualized by a high content screening (HCS). Instrument parameter adjustment and image capture were performed using Harmony 4.9 software, which comes with the HCS. The image acquisition mode was confocal; a 63 × Water. NA 1.15 objective was used. The acquisition heights (*Z*-axis) were 2.5 μm, 3 μm, and 3.5 μm, and the target fluorescence color was adjusted to pure red, the Hoechst channel was pure blue, and the DiD channel was pure green.

### 2.9. Pharmacokinetic Analysis

Liposomes and free drugs were all dissolved and diluted to the required concentration with 10% sucrose.

CD-1 female mice were randomly divided into four groups, with 18 or 24 mice in each group. Mice were intravenously injected with Lipo-1 (10:1), Lipo-2 (10:1), Lipo-3 (10:1), or combined free drugs. After the administration, mice injected with liposomes were euthanized. Blood taken from the heart and femoral bone marrow was collected at 15 min, 1 h, 4 h, 8 h, 24 h, and 48 h. Mice injected with free drugs were euthanized. Blood taken from the heart and femoral bone marrow was collected at 5 min, 15 min, 30 min, 1 h, 2 h, 4 h, 8 h, and 24 h. Drug content (ng/femur) in the femoral bone marrow was calculated based on the theoretical volume of mouse femoral bone marrow being 8.9 µL/femur.

Three male SD rats were used to evaluate the PK parameters of Lipo-5 (30:1). Rats were intravenous injected with Lipo-5, and after the administration, the blood was taken from the jugular vein at 5 min, 15 min, 30 min, 1 h, 2 h, 4 h, 8 h, 12 h, 24 h, and 48 h.

The samples collected from blood and femoral bone marrow were determined by LC-MS/MS to evaluate the concentration of cytarabine and idarubicin in liposomes. Phoenix WinNonlin software (version 8.3) was used to calculate pharmacokinetic parameters according to the non-compartmental model, including the area under the drug-time curve (AUC_0-t_ and AUC_0-∞_), time to peak (T_max_), peak concentration (C_max_), elimination half-life (t_1/2_), mean residence time (MRT), clearance rate (CL), and steady-state apparent volume of distribution (V_ss_).

### 2.10. In Vivo Antitumor Effect

Liposomes and free drugs were all dissolved and diluted to the required concentration with a 10% sucrose solution.

To determine the MTD dose of liposomes and combined free drugs, WEHI-3B cells (1E6) were intraperitoneally injected into the CD-1 nude female mice, which were randomly divided into nine groups with five mice in the liposomes group and three mice in the free drugs group. Mice were intravenous injected with the liposomes or combined free drugs at days 1, 4, and 7; details of the grouping are shown in Table 2. During the experiment, mice were observed once a day, and body weight, clinical manifestation, and death were recorded.

As for the efficacy experiment, WEHI-3B cells (1E6) were intraperitoneally injected into the CD-1 nude female mice; mice were randomly divided into nine groups with six mice in each group. Mice were intravenously injected with the vehicle (10% sucrose), liposomes, or combined free drugs at days 1, 4, and 7; details of the grouping are shown in Table 3. During the experiment, mice were observed once a day. Observations included at least death (time of death, pre-death reactions, etc.), skin, hair, eyes, ears, nose, behavior, appearance, etc. The animals were weighed and recorded every afternoon during the experiment.

### 2.11. Cell Cycle Analysis

Kasumi-1 cells were seeded in 96-well plates, in an appropriate complete cell culture medium. Cells were treated with the combination of two free drugs for 24 h, the concentration of cytarabine was fixed (3.33 μM) and the concentration of idarubicin was adjusted to different molar ratios (5:1, 10:1, 20:1, 30:1, 40:1, 50:1). Both cytarabine and idarubicin dissolved in DMSO, and DMSO was used as the negative control with maximal concentration of 0.4% (*v*/*v*). After incubation, cells were centrifuged at 1000× *g* for 5 min, washed with 200 μL of cooled PBS for one time, and then 200 μL of cooled 70% ethanol was added with gentle mixing to fix the cells at 4 °C for 4 h. After fixing, cells were centrifuged at 1000× *g* for 5 min, washed with 200 μL of cooled PBS again, and then 200 μL of propidium iodide staining solution was added to each sample with slowly and fully resuspending. Cells were incubated at 37 °C for 30 min, after that flow cytometry was completed immediately by using a Beckman FlowFlex flow cytometer. For the analysis of the cell cycle, the original data was fitted by using the cell cycle fitting software ModFitLT (Version 5.0).

### 2.12. RNA Isolation and Real-Time Quantitative RT-PCR

Kasumi-1 cells were seeded in 12-well plates and treated with a combination of two free drugs for 24 h; the concentration of cytarabine was fixed (3.33 μM), and the concentration of idarubicin was adjusted to different molar ratios (5:1, 10:1, 20:1, 30:1, 40:1, 50:1). Both cytarabine and idarubicin dissolved in DMSO, and DMSO was used as the negative control, with a maximal concentration of 0.5% (*v*/*v*). After incubation, cells were collected, and the total RNA from them was isolated using a FastPure cell/tissue total RNA isolation kit. The cDNA was synthesized using 1 μg of total RNA and reverse transcribed using a First-Strand cDNA Synthesis SuperMix. Gene expression was measured through real-time PCR using gene-specific primers and 1 × Taq Pro Universal SYBR qPCR Master Mix in a total volume of 20 μL. The PCR reactions were performed on a Light Cycler 480 II (Roche, Mannheim, Germany) using a thermal profile of 10 min at 95 °C, followed by 40 cycles of 15 s at 95 °C, 60 s at 1 min, heating to 95 °C, and cooling for 30 s at 4 °C. The results were analyzed using LightCycler 480 software (Roche, Mannheim, Germany). The relative levels of mRNA were analyzed using the △△Ct method. The primers were designed and synthesized at General Biol (Chuzhou, China). The primer sequences are shown in Table 4.

### 2.13. Statistical Analysis

The inhibition rate (%) = ×100%, and IC50 (half maximal inhibitory concentration) was defined as the drug concentration when the inhibitory effect on cell proliferation reached 50% of the normal cell proliferation level, which was calculated using GraphPad Prism 7 for Windows (GraphPad Software, La Jolla, CA, USA). The combination index (CI) was calculated by CompuCyn (Version 1.0) [36]. For the experiment of PK, Phoenix WinNonlin was used to calculate the PK parameters. For the analysis of the cell cycle, the original data were fitted using the cell cycle fitting software ModFitLT (Version 5.0). For an in vivo experiment, data were calculated and statistically processed using Microsoft Excel 2007 software. For the cell cycle and gene expression assay results, data were analyzed using SPSS software (Version 16.0). One-way ANOVA was applied for multiple comparisons, with LSD post hoc analysis for data meeting homogeneity of variance or with Dunnett’s T3 analysis for data not assuming equal variances. Data were expressed as mean ± standard error (Mean ± SE) unless otherwise specified.

## 3. Results and Discussion

### 3.1. Screening of Synergistic Molar Ratios of Cytarabine and Idarubicin in Cell Lines

Since cytarabine and idarubicin are used separately in the inductive treatment for acute leukemia, such an administration method and dose may not be the best choice. To find the best combined ratio of the two drugs, we selected nine leukemia cell lines (KG-1, OCI-AML-3, Kasumi-1, HL-60, MV-4-11, WEHI-3B, Molt-4, CCRF-CEM, MOLM-13) and four solid tumor cell lines (HCT116, KP-4, TOV-21G, ES-2) to perform the screening of the best combination molar ratio. The molar ratios of cytarabine and idarubicin were set at 5:1, 10:1, 20:1, 30:1, 40:1, and 50:1, and cells were incubated with combined drugs for 72 h at different molar ratios. After incubation, cell viability was detected by CTG assay, and the CI value was calculated. The fraction affected (Fa) value was defined as the fraction of cell viability affected. The CI values of 13 cell lines at different molar ratios when the Fa value was 0.9 are shown in Table 5. The sensitivity of each cell line to varied molar ratios of cytarabine and idarubicin was different. Specifically, when the Fa value was 0.9 and the molar ratio of cytarabine to idarubicin was 30:1, the proportion of cells showing synergistic effect was the highest (80%) in all cell lines, which was lowest at 5:1, and the results at 10:1, 20:1, 40:1, and 50:1 were similar (Figure 1A). Among all leukemia cells, the overall results were similar to those for all cells, with the highest proportion of cells showing synergism at a molar ratio of 30:1, followed by a molar ratio of 20:1 (Figure 1B). CompuSyn is a computer program for the quantitation of synergism and antagonism in drug combinations and the determination of IC50 and ED50 values; the reliability of this software has been demonstrated in numerous previous studies [37,38,39,40]. Consequently, the molar ratio of the optimal synergistic effect of cytarabine and idarubicin as calculated using this approach may offer valuable insights.

### 3.2. Preparation and Characterization of Liposomes

The ion gradient is usually used during the active-loading method, such as ammonium sulfate, transit metal ions [33,41], etc. In this study, we evaluated different ion gradients for the entrapment of idarubicin as the formulations of Lipo-1, Lipo-2, and Lipo-3 (see Table 4). A molar ratio of 10:1 was used as it was easy to control and helpful to screen for qualified ion gradients in our early study. The characteristic parameters of these three formulations, such as the particle size, zeta potential, and EE% of cytarabine and idarubicin, displayed no significant difference. These results suggested that the ammonium sulfate and transit metal ions (Cu^2+^, Fe^2+^) may effectively entrap idarubicin. The same result was observed in the in vitro stability study, suggesting a stable liposome formulation was formed. Therefore, the subsequent in vivo PK studies were used for the screening of liposome formulations with different ion gradients.

As the results of cell lines showed, molar ratios of cytarabine/idarubicin of 20:1~40:1 were demonstrated to be synergistic efficiency. Furthermore, to screen the optimal molar ratio of cytarabine/idarubicin, liposome formulations at molar ratios of 20:1, 30:1, and 40:1 were prepared (see Table 6). The characteristic parameters showed that the sizes of liposomes were controlled by extrusion to 80~120 nm. In the sequential method, the particle size was controlled after loading cytarabine and was influenced by extrusion and the subsequent drug encapsulation. The final particle size of the Lipo-1 to Lipo-6 was approximately 100 nm with PDI ≤ 0.1, respectively, which suggested an acceptable particle size distribution with adequate homogeneity and potential to reduce interactions with plasmatic proteins when injected by the intravenous route. Liposomes with particle sizes ≤120 nm can escape from the recognition and phagocytosis of the mononuclear phagocytic system [42,43]. The zeta potentials of these formulations were approximately −30 mV, attributed to the negatively charged DSPG, which keeps the liposomes repellent to cells, further leading to longer circulation and good stability in plasma and a higher accumulation rate in tumor tissue [44]. The typical particle size distributions and zeta potentials for the liposomes are shown in Figure 2A.

The EE of cytarabine and idarubicin is shown in Table 6, all of which were greater than 95%, making it feasible for potential drug delivery applications. The final EE of cytarabine was nearly 100%, as the free cytarabine was removed with the tangential flow filtration method, although a passive-loading method was used. Meanwhile, the high EE of cytarabine also suggested the stability of cytarabine during the subsequent loading process of idarubicin, with minimal or almost no leakage. For idarubicin, data evidenced that there was no significant difference in the encapsulation of idarubicin among liposomes with different molar ratios, which were all close to 100%, suggesting a possible unsaturation of the bilayer. On the other hand, no significant difference was observed after cytarabine encapsulation for any of the formulations, demonstrating that cytarabine co-encapsulation did not alter the ability of the liposomal system to load idarubicin. These results allow the prescription to be used for the preparation of liposomes with different cytarabine/idarubicin molar ratios.

Figure 2B shows the morphological imaging measured by cryo-TEM. It shows a spherical structure and the uniform distribution of liposomes with 80 nm particle sizes.

### 3.3. In Vitro Stability

The hydrolysis of acyl chains and the oxidation of unsaturated bonds in phospholipids cause lipid membrane fusion, aggregation, and drug leakage. To provide stable liposome formulations for subsequent in vitro and in vivo studies, the long-term stability of liposomes was studied at 4 °C for 21 days. The particle size, ZP, and EE of liposomes were monitored for 21 days and stored at 4 °C. There was no formation of aggregates observed in the liposomes stored at 4 °C within 21 days. The particle size showed a slight increment, while the ZP slightly decreased. The EE showed no significant change within 21 days. Therefore, the liposome formulations were relatively stable after 21 days when stored at 4 °C (see Table 7), which may be attributed to the thermodynamic and physicochemical stability of the DSPC/DSPG/Chol (7/2/1, mol%) system [13] and the interactions between drugs and copper ions [13].

### 3.4. In Vitro Cell Uptake and Cytotoxicity of Liposomes

We evaluated cell uptake of cytarabine-idarubicin liposome by inhibition of cell proliferation and observation of the fluorescence intensity of idarubicin inside cells. CCRF-CEM cells were incubated with a gradient concentration of Lipo-5 for 72 h, and the combination of free drugs in the same molar ratio was used as the control. The results showed that the IC50 of the free cytarabine-idarubicin combination on CCRF-CEM cells was 216.12 nM, while the inhibitory effect of liposomes on the cells was significantly weaker than that of free drugs with an IC50 value of 5826 nM (Table 8, Figure 3A,B), because the drugs in the liposomes must be released gradually rather than directly interacting with cells to exert their killing effect.

We further tested the intracellular uptake of liposomes by examining the intracellular accumulation of idarubicin (by virtue of its inherent fluorescence) using high-content screening. CCRF-CEM cells were incubated with Lipo-1 for up to 16 h and imaged for idarubicin fluorescence. As the incubation time extended, the cells treated with Lipo-1 exhibited a gradual death, as evidenced by a reduction in the number of viable cells within the brightfield (BF) and a concomitant decrease in the number of nuclei, which displayed gradual shrinking and condensation (Hoechst). The images also showed that idarubicin (shown as red) accumulated within cells as early as 0.5 h, and the intensity of fluorescence decreased rapidly with the extension of time, mainly because of the cell death (Figure 4). Before high-content screening, cells should be washed three times, which can discard most of the cell fragments containing liposomes. Therefore, intracellular idarubicin fluorescence is indicative of liposome uptake by the leukemia cells, consistent with the appearance of intracellular fluorescence.

### 3.5. Pharmacokinetic

To verify whether cytarabine-idarubicin liposomes can maintain a constant molar ratio in vivo, we first used liposomes (Lipo-1, Lipo-2, and Lipo-3) with a relatively stable molar ratio of 10:1 for PK detection. The drug-carrying ions were different among Lipo-1, Lipo-2, and Lipo-3 as Glu-Cu^2+^, Glu-Fe^2+^, and (NH_4_)_2_SO_4_, therefore the PK parameters of these three liposomes were also compared. The results showed that free cytarabine was rapidly metabolized in vivo, with the blood concentration reduced at least 1000 times within 4 h, while Lipo-1 significantly slowed down the release of cytarabine in plasma (Figure 5A). The dose of idarubicin in liposomes was slightly higher than idarubicin in combined free drugs; however, the plasma concentration of idarubicin in mice given liposomes was significantly higher than those given combined free drugs (Figure 5A). At the same time, we detected the concentration of cytarabine and idarubicin within bone marrow after the injection of liposomes or free drugs, and the results were basically consistent with those in plasma. Cytarabine in combined free drugs showed PK characteristics of high initial concentration and extremely fast metabolism, while liposomes could delay the release and metabolism of cytarabine in bone marrow (Figure 5B). However, the concentrations of idarubicin in the bone marrow of mice given liposomes or combined free drugs were more similar, which was different from that in plasma (Figure 5B). The PK characteristics of liposomes in plasma and bone marrow were not significantly improved after the replacement of different drug-loading ions (Figure 5C–F). We calculated the molar ratio of cytarabine to idarubicin in plasma and bone marrow after the injection of three liposomes or combined free drugs. The results showed that the three liposomes could basically maintain a molar ratio of 10:1 in plasma for 8 h, among which Lipo-1 was the most stable (Figure 5G). However, the molar ratio dropped sharply with the rapid decrease in cytarabine within the combined free drugs (Figure 5H). In the bone marrow, Lipo-1 could maintain a molar ratio of about 10:1 within 24 h, while Lipo-2 and Lipo-3 fluctuated more significantly (Figure 5I). After the injection of the combined free drugs, the change in the molar ratio of cytarabine and idarubicin in bone marrow was similar to that in plasma (Figure 5J). A comparison of the PK behavior of cytarabine in mouse plasma and bone marrow in the literature revealed that the data and trends were essentially consistent with those obtained in this study, thereby substantiating the reliability of the system employed in this study [6]. According to the analysis of the above results based on PK parameters, the most fundamental reason why Lipo-1 is superior to Lipo-2 and Lipo-3 in the maintenance of the molar ratio is that the clearance rate of idarubicin in Lipo-1 is lower than that of the other two liposomes, whether in plasma or in bone marrow (Table 9). According to the PK diagnosis of the two drugs, it can also be inferred that delaying the clearance of idarubicin in vivo is the key factor in maintaining the stability of the molar ratio of cytarabine and idarubicin. Therefore, Glu-Cu^2+^ was subsequently used as the drug-loading ion to prepare liposomes.

Since the molar ratio of cytarabine to idarubicin is 30:1, the proportion of cells showing synergistic effects was highest. We then investigated the plasma PK characteristics of Lipo-5 (drug-loading ion as Glu-Cu^2+^) in SD rats. The results showed that after the injection of Lipo-5 or combined free drugs, the PK characteristics of cytarabine and idarubicin in rat plasma were basically the same as those in mice (Figure 5A and Appendix A), and the molar ratio of cytarabine to idarubicin in plasma was maintained at 30:1 within 4 h (Appendix A).

### 3.6. In Vivo Therapeutic Efficacy and Toxicity

Before conducting in vivo efficacy studies in mice, we first explored the maximum tolerated dose (MTD) of liposomes and free drug combinations in CD-1 nude mice. Lipo-4 was chosen, and mice were intravenously injected with Lipo-4 or combined free drugs on days 1, 4, and 7. The body weight changes, clinical manifestations, and death of mice were recorded. The MTD dose was determined as the dose with a loss of body weight of no more than 15% within 14 days after the last dose, and the duration of weight at the lowest point was ≤2 days. The results showed that when the dose of Lipo-4 was 5 mg/kg (as measured by cytarabine), the weight of mice was maintained well, and there was a slight increase after the last dose. When the dose of Lipo-4 was more than 10 mg/kg (as measured by cytarabine), the weight of mice was significantly decreased. Thus, the MTD dose of Lipo-4 was between 5 mg/kg and 10 mg/kg (as measured by cytarabine) (Figure 6A,B). The MTD of mice administered with the free drug combination was 200:1.0 or 300:0.9 (dose of cytarabine: idarubicin) (Figure 6A,B). We can see that the dose-limiting factor that affected the weight of mice was idarubicin. Therefore, we set the dosage of idarubicin in liposomes as 0.55 mg/kg and 0.77 mg/kg in subsequent experiments, and the MTD dose of the free drug combination as 300:0.9 (dose of cytarabine: idarubicin).

In a WEHI-3B mouse model with ascites syndrome, we simultaneously compared the efficacy of cytarabine-idarubicin liposomes with a molar ratio of 20:1 (Lipo-4), 30:1 (Lipo-5), 40:1 (Lipo-6), and two doses of free drug combinations. Mice were intravenously injected with liposomes or combined free drugs on days 1, 4, and 7, and the weight changes and survival time were recorded. The results showed the body weight gradually increased independent of the formulations or free drug, indicating that the dose of liposomes and free drugs was well tolerated. At the same time, we found that the change in body weight was positively correlated with the volume of abdominal water, which means a significant increase in body weight indicating a poor treatment effect (Figure 6C). In terms of efficacy, the longest survival time of mice in the control group was only 22 days, which could be slightly prolonged by the free drug combination (5:0.55). When the MTD dose of the free drug combination was given, the survival time was extended to 35 days. Compared with the combined free drugs, the efficacy of liposomes was significantly enhanced, and when the molar ratio was 30:1, mice had the longest survival time of 42 days, followed by 20:1 and 40:1. The results were highly consistent with those obtained from in vitro cell lines (Figure 6D). WEHI-3B cells, a type of mouse myeloid leukemia cell, have been utilized in previous investigations to assess the in vivo antitumor efficacy of anti-leukemia drugs [6]. Furthermore, this model is highly reliable. In this study, the survival time of the control group aligned with that observed in previous literature reports [6], and the data produced from our repeated experiments was highly consistent. Therefore, this model can better represent the anti-AML efficacy of liposomal or free drugs.

### 3.7. Cell Cycle

To investigate the mechanism of the cell killing, Kasumi-1 cells were treated with a combination of two free drugs. The concentration of cytarabine was fixed at 3.33 μM, and the concentration of idarubicin was adjusted with different molar ratios from 5:1 to 50:1. The results showed that the proportion of cells in the S phase increased rapidly from 20:1 to 40:1 (Figure 7A,B). The specific changes in the cell cycle are shown in Figure 7C–N.

To further investigate how cytarabine and idarubicin synergistically regulate the cell cycle, we examined genes associated with the cell cycle. The results showed that when the molar ratio of cytarabine/idarubicin was in the range of 20:1~30:1, its upregulation of *CDK1*, *CDK4*, *CCNA2*, *CCNB1*, *CCND1*, *CCND2*, and *CCNE1* genes in the cells was significantly stronger than that of the control group and other treatment groups (the molar ratios of cytarabine/idarubicin were 5:1, 10:1, 40:1, and 50:1) (Figure 8).

Among these upregulated genes, the upregulation of *CDK4*, *CCNA2*, *CCND1*, *CCDN2*, and *CCNE1* genes can promote the transition of cells from the G1 phase to the S phase. Idarubicin is an anthracycline antibiotic that exerts its antitumor effect mainly by inhibiting topoisomerase II and interfering with DNA synthesis, so it mainly affects the S phase. Cytarabine is a pyrimidine anti-metabolite drug that simulates normal cytosine nucleosides and is incorporated into the DNA chain, resulting in the blockage of the function of DNA polymerase, thereby inhibiting DNA synthesis and replication. In addition, cytarabine is particularly sensitive to cells in the S phase, so it mainly affects the S phase of the cell cycle. In summary, the combination of cytarabine and idarubicin with a specific molar ratio (20:1 or 30:1) can upregulate the expression of multiple genes, causing more cells to stagnate in the S phase, so that cytarabine and idarubicin can better exert their cell-killing effects.

## 4. Conclusions

In this work, dual-loaded liposomes containing cytarabine and idarubicin were formulated and their physiochemical and pharmacologic properties were evaluated.

The optimal molar ratio interval for the synergistic effects of cytarabine and idarubicin was determined through adequate in vitro cellular studies. Accordingly, liposomes of cytarabine and idarubicin were formulated in a range of prescriptions and molar ratios, and a systematic investigation was conducted into their physicochemical properties, stabilization, antitumor efficacy, and mechanism of action. The physicochemical properties of the prepared liposomes were verified to be favorable, exhibiting stability in vitro. Moreover, the cellular uptake of the drugs from liposomes was confirmed, as was the superior drug molar ratio maintenance in rodent plasma and bone marrow exhibited by the liposomes when compared to the free drugs. Additionally, in vivo antitumor experiments demonstrated the efficacy superiority of the dual-drug liposomes over free drug combinations, along with the optimal molar ratio of cytarabine to idarubicin. The preliminary clarification of the pharmacological mechanism underlying the synergistic effects of the two drugs was also achieved.

However, the time of both drugs maintained in the plasma and bone marrow was shorter than Vyxeos^TM^, and their molar ratios were also less, due to the fast release of idarubicin. To improve these disadvantages, we plan to modify liposomes with long-cycle materials (such as PEG) or targeted groups to further prolong the retention time of liposomes in blood and bone marrow, thereby enhancing efficacy.

## Figures and Tables

**Figure 1 pharmaceutics-16-01220-f001:**
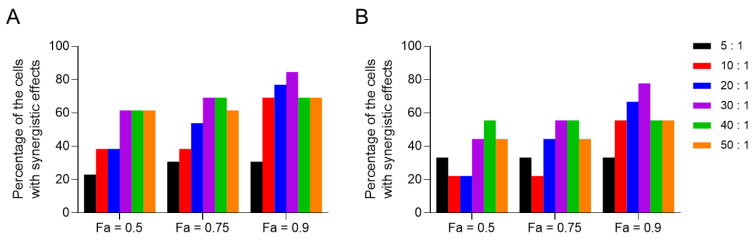
The proportion of cells with different molar ratios of cytarabine and idarubicin exerting synergistic effects**.** (**A**) When Fa values were 0.5, 0.75, and 0.9, the ratio of cytarabine to idarubicin was set at 5:1, 10:1, 20:1, 30:1, 40:1, and 50:1, the proportion of all 13 cell lines showing synergistic effect (CI value less than 1). (**B**) When Fa values were 0.5, 0.75, and 0.9, the ratio of cytarabine to idarubicin was set at 5:1, 10:1, 20:1, 30:1, 40:1, and 50:1, the proportion of nine leukemia cells showing synergistic effect (CI value less than 1).

**Figure 2 pharmaceutics-16-01220-f002:**
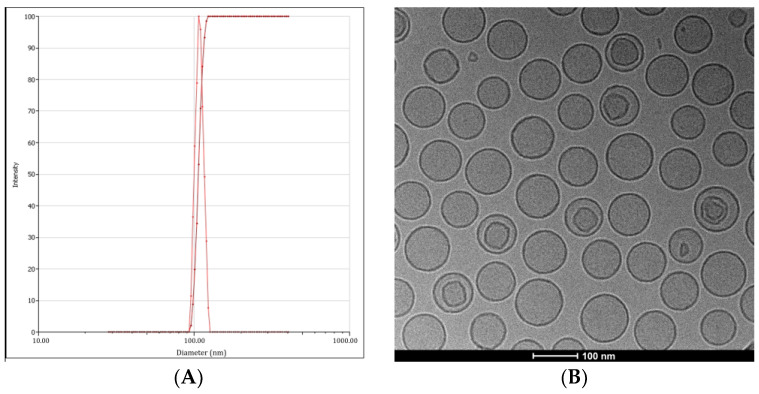
Typical profile of the dual drug liposome with a cytarabine/idarubicin mole ratio of 30:1 with Glu-Cu^2+^ as a carrier ion. (**A**): Diagram of intensity-particle size distribution. The peak indicates the particle size distribution, the curve indicates the cumulative percentage. (**B**): Diagram of Cyro-TEM.

**Figure 3 pharmaceutics-16-01220-f003:**
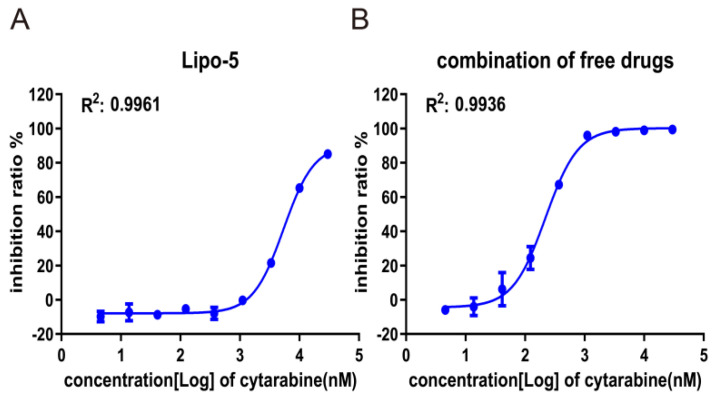
In vitro cell uptake of cytarabine-idarubicin liposome on CCRF-CEM cells**.** (**A**) The proliferation inhibition curve of CCRF-CEM cells treated with Lipo-5; Lipo-5 was diluted three times from 30,000 nM to 4.6 nM (measured by cytarabine concentration), and the experiment was repeated three times. (**B**) The proliferation inhibition curve of CCRF-CEM cells treated with combined free drugs of cytarabine-idarubicin (molar ratio at 30:1); free drugs were diluted three times from 30,000 nM to 4.6 nM (measured by cytarabine concentration), and the experiment was repeated three times.

**Figure 4 pharmaceutics-16-01220-f004:**
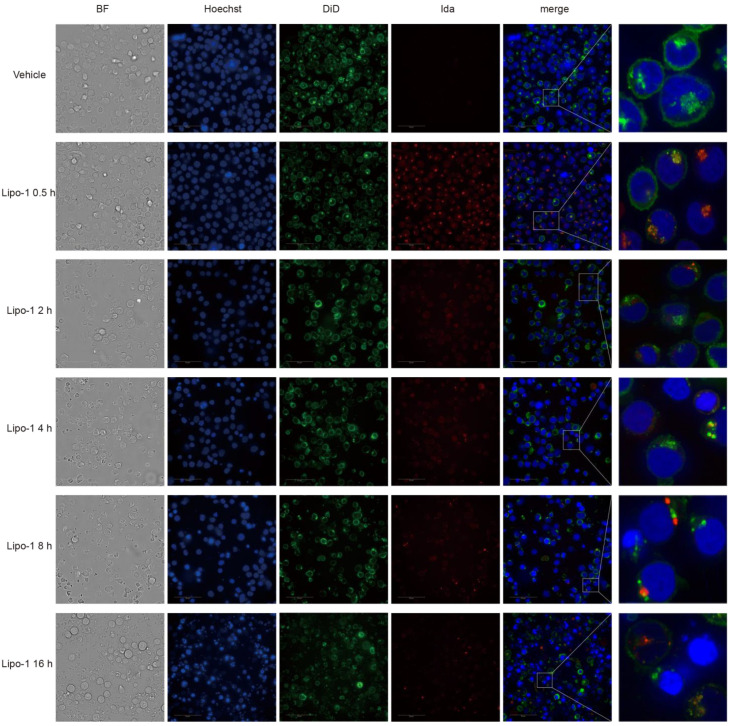
In vitro cell uptake of cytarabine-idarubicin liposome on CCRF-CEM cells**.** The cell membrane was stained with DiD for 15 min, and then the dyestuff was removed by centrifugation, and CCRF-CEM cells were incubated with Lipo-1 (500 μM) for 0.5 h, 2 h, 4 h, 8 h, and 16 h. After incubation, nuclei were stained with Hoechst for 0.5 h, and then cells were centrifuged to remove dyestuff. After that, cells were washed three times with cold HBSS and visualized using a high-content screening (HCS).

**Figure 5 pharmaceutics-16-01220-f005:**
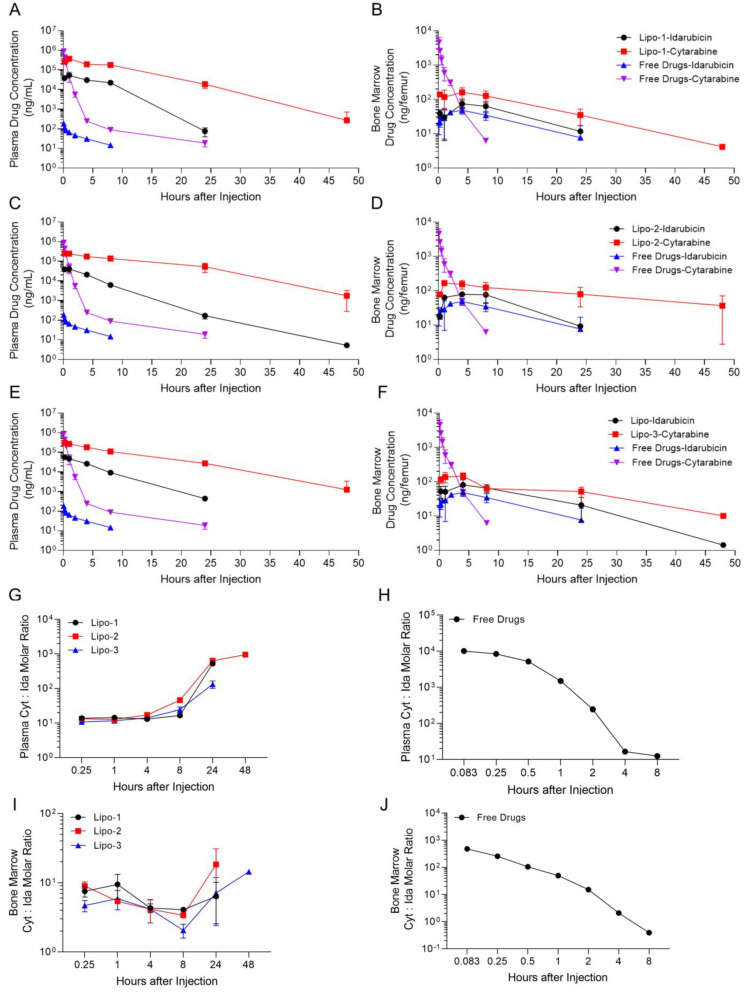
PK characteristics of CD-1 mice treated with cytarabine-idarubicin liposome. CD-1 mice were intravenously injected with three kinds of prescription cytarabine-idarubicin liposomes (the molar ratio of cytarabine to idarubicin was 10:1, the dose of cytarabine was 12 mg/kg, and the dose of idarubicin was 2.63 mg/kg) or the combination of cytarabine-idarubicin free drugs (the dose of cytarabine was 600 mg/kg, and the dose of idarubicin was 1.8 mg/kg). The plasma and bone marrow of mice were collected at different time points to detect the concentrations of cytarabine and idarubicin; three samples were collected at each time point. (**A**) Drug concentration-time curves of cytarabine and idarubicin within plasma after injection of Lipo-1 and combined free drugs in CD-1 mice. (**B**) Drug concentration-time curves of cytarabine and idarubicin within bone marrow after injection of Lipo-1 and combined free drugs in CD-1 mice. (**C**) Drug concentration-time curves of cytarabine and idarubicin within plasma after injection of Lipo-2 and combined free drugs in CD-1 mice. (**D**) Drug concentration-time curves of cytarabine and idarubicin within bone marrow after injection of Lipo-2 and combined free drugs in CD-1 mice. (**E**) Drug concentration-time curves of cytarabine and idarubicin within plasma after injection of Lipo-3 and combined free drugs in CD-1 mice. (**F**) Drug concentration-time curves of cytarabine and idarubicin within bone marrow after injection of Lipo-3 and combined free drugs in CD-1 mice. (**G**) The molar ratio changes of cytarabine to idarubicin in plasma at different time points in CD-1 mice after intravenous administration of Lipo-1, Lipo-2, and Lipo-3. (**H**) The molar ratio changes of cytarabine to idarubicin in plasma at different time points in CD-1 mice after intravenous administration of combined free drugs. (**I**) The molar ratio changes of cytarabine to idarubicin in bone marrow at different time points in CD-1 mice after intravenous administration of Lipo-1, Lipo-2, and Lipo-3. (**J**) The molar ratio changes of cytarabine to idarubicin in bone marrow at different time points in CD-1 mice after intravenous administration of combined free drugs.

**Figure 6 pharmaceutics-16-01220-f006:**
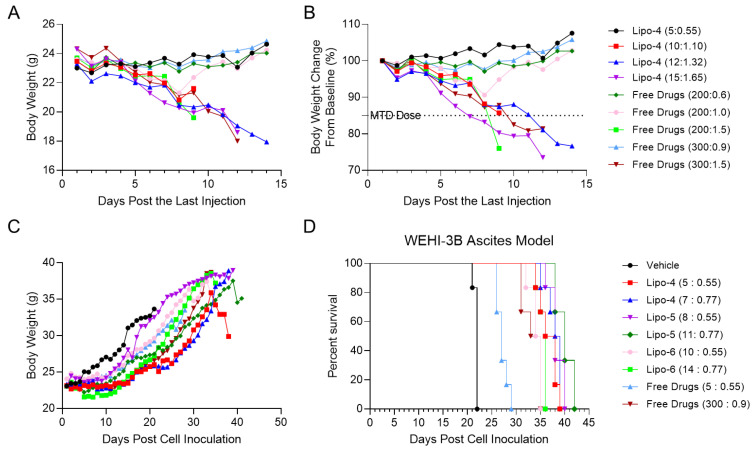
Efficacy of cytarabine-idarubicin liposomes in WEHI-3B ascites model. Female CD-1 nude mice were intravenously injected with Lipo-4 as 5 mg/kg, 10 mg/kg, 12 mg/kg, 15 mg/kg (as measured by cytarabine), free drug combinations as 200:0.6 mg/kg, 200:1.0 mg/kg, 200:1.5 mg/kg, 300:0.9 mg/kg, and 300:1.5 mg/kg on D1, D4, and D7 to explore the maximum tolerated dose of the liposome and free drug combination. (**A**) Weight change curves of CD-1 nude mice within 14 days after the last dose, with five animals per group. (**B**) Percentage change in body weight from baseline within 14 days after the last dose, with five animals per group. Female CD-1 nude mice were intravenously injected with Lipo-4, Lipo-5, Lipo-6, and free drug combinations on D1, D4, and D7 after the WEHI-3B ascites model was set successfully. (**C**) The weight change curve of each group of mice after modeling, with six animals per group. (**D**) Survival rate curves of mice in each group after modeling, with six animals per group.

**Figure 7 pharmaceutics-16-01220-f007:**
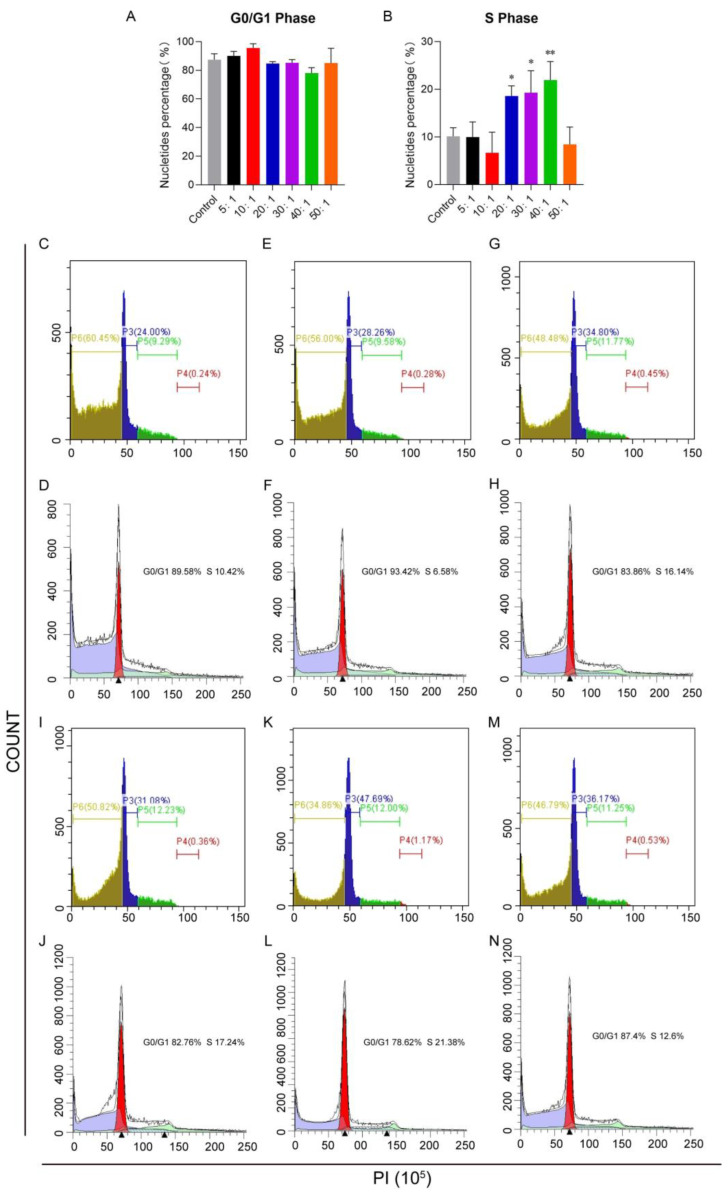
Cell cycle analysis of Kasumi-1 cells treated with combined free drugs. Kasumi-1 cells were treated with combined free drugs to explore their effects on the cell cycle. (**A**) The proportion of cells blocked on the G0/G1 phase. (**B**) The proportion of cells blocked on the S phase. (**C**,**D**) cell cycle analysis as molar ratio was 5:1, (**E**,**F**) cell cycle analysis as molar ratio was 10:1, (**G**,**H**) cell cycle analysis as molar ratio was 20:1, (**I**,**J**) cell cycle analysis as molar ratio was 30:1, (**K**,**L**) cell cycle analysis as molar ratio was 40:1, (**M**,**N**) cell cycle analysis as molar ratio was 50:1. Data were shown as mean ± SD, * *p* < 0.05, ** *p* < 0.01, compared with the control group (**A**,**B**). The experiment was repeated four times.

**Figure 8 pharmaceutics-16-01220-f008:**
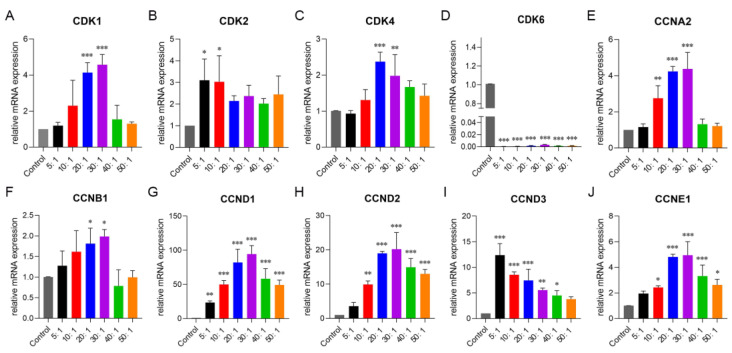
Gene expression was associated with the cell cycle in Kasumi-1 cells as the concentration of cytarabine was 3.33 μM. Kasumi-1 cells were treated with combined free drugs to explore its effects on gene expression associated with the cycle cell cycle. Gene expression of CDK1 (**A**), CDK2 (**B**), CDK4 (**C**), CDK6 (**D**), CCNA2 (**E**), CCNB1 (**F**), CCND1 (**G**), CCND2 (**H**), CCND3 (**I**) and CCNE1 (**J**) was measured by qRT-PCR. Data were shown as mean ± SD, * *p* < 0.05, ** *p* < 0.01, *** *p* < 0.001, compared with the control group. The experiment was repeated three times.

**Table 1 pharmaceutics-16-01220-t001:** Cells used in this experiment.

Cell Lines	Cells per Well	Cell Culture Media
OCI-AML-3	15,000	RPMI-1640 medium plus 20% fetal bovine serum
KG-1	11,000	IMDM medium plus 20% fetal bovine serum
HL-60	10,000	RPMI-1640 medium plus 10% fetal bovine serum
Kasumi-1	10,000	RPMI-1640 medium plus 20% fetal bovine serum
MV-4-11	8000	IMDM medium plus 10% fetal bovine serum
CCRF-CEM	8000	RPMI-1640 medium plus 10% fetal bovine serum
MOLM-13	8000	RPMI-1640 medium plus 20% fetal bovine serum
Molt-4	7500	RPMI-1640 medium plus 10% fetal bovine serum
WEHI-3B	7500	IMDM medium plus 10% fetal bovine serum, 0.05 nM 2-mercaptoethanol
TOV-21G	2000	RPMI-1640 medium plus 15% fetal bovine serum
KP-4	2000	RPMI-1640 medium plus 10% fetal bovine serum
HCT116	1500	RPMI-1640 medium plus 10% fetal bovine serum
ES-2	1000	McCoy’s 5A medium plus 10% fetal bovine serum

**Table 2 pharmaceutics-16-01220-t002:** Grouping details for MTD dose-finding experiments of liposomes and combined free drugs in CD-1 nude mice.

Testing Drugs	Dose of Cytarabine (mg/kg)	Dose of Idarubicin (mg/kg)
Lipo-4	5	0.55
Lipo-4	10	1.10
Lipo-4	12	1.32
Lipo-4	15	1.65
Free drugs	200	0.6
Free drugs	200	1.0
Free drugs	200	1.5
Free drugs	300	0.9
Free drugs	300	1.5

**Table 3 pharmaceutics-16-01220-t003:** Grouping details for in vivo efficacy (the WEHI-3B model) of liposomes and combined free drugs in CD-1 nude mice.

Testing Drugs	Dose of Cytarabine (mg/kg)	Dose of Idarubicin (mg/kg)	Molar Ratio of Cytarabine and Idarubicin
Vehicle (10% sucrose)	0	0	NA
Lipo-4	5	0.55	20:1
Lipo-4	7	0.77	20:1
Lipo-5	8	0.55	30:1
Lipo-5	11	0.77	30:1
Lipo-6	10	0.55	40:1
Lipo-6	14	0.77	40:1
Free drugs	5	0.55	20:1
Free drugs	300	0.9	730:1

**Table 4 pharmaceutics-16-01220-t004:** Primer sequences of the genes.

Genes	Species	Primer Sequences (5′ to 3′)	
		Forward	Reverse
*CCNA2*	Homo sapiens	GGATGGTAGTTTTGAGTCACCAC	CACGAGGATAGCTCTCATACTGT
*CCNB1*	Homo sapiens	AATAAGGCGAGATCAACATGGC	TTTGTTACCAATGTCCCCAAGAG
*CCND1*	Homo sapiens	GCTGCGAAGTGGAAACCATC	CCTCCTTCTGCACACATTTGAA
*CCND2*	Homo sapiens	ACCTTCCGCAGTGCTCCTA	CCCAGCCAAGAAACGGTCC
*CCND3*	Homo sapiens	TACCCGCCATCCATGATCG	AGGCAGTCCACTTCAGTGC
*CCNE1*	Homo sapiens	GCCAGCCTTGGGACAATAATG	CTTGCACGTTGAGTTTGGGT
*CDK1*	Homo sapiens	AAACTACAGGTCAAGTGGTAGCC	TCCTGCATAAGCACATCCTGA
*CDK2*	Homo sapiens	CCAGGAGTTACTTCTATGCCTGA	TTCATCCAGGGGAGGTACAAC
*CDK4*	Homo sapiens	ATGGCTACCTCTCGATATGAGC	CATTGGGGACTCTCACACTCT
*CDK6*	Homo sapiens	GCTGACCAGCAGTACGAATG	GCACACATCAAACAACCTGACC
*GAPDH*	Homo sapiens	TGCACCACCAACTGCTTA	GGATGCAGGGATGATGTTC

**Table 5 pharmaceutics-16-01220-t005:** CI values of cell lines at different molar ratios of cytarabine and idarubicin when the Fa value is 0.9.

Cell Lines	Tumor Types	Molar Ratio (Cytarabine: Idarubicin)					
		5:1	10:1	20:1	30:1	40:1	50:1
KG-1	Human acute myeloid leukemia	0.80	1.03	0.67	0.53	0.76	0.69
OCI-AML-3	Human acute myeloid leukemia	0.50	0.80	0.73	0.74	0.86	0.93
Kasumi-1	Human acute myeloblastic leukemia	2.58	1.32	1.06	0.83	0.61	0.55
HL-60	Human acute promyelocytic leukemia	0.83	0.69	0.71	0.69	1.00	0.83
MV-4-11	Human myeloid leukemia	1.30	0.96	0.92	0.78	1.02	1.07
WEHI-3B	Murine myeloid leukemia	1.08	0.99	1.11	1.64	1.31	2.69
Molt-4	Human acute lymphocytic leukemia	1.12	0.84	0.70	0.84	0.75	0.54
CCRF-CEM	Human acute lymphocytic leukemia	1.13	0.86	0.78	0.76	0.85	0.82
MOLM-13	Human acute myeloid leukemia	0.94	0.82	0.88	1.28	1.54	2.18
HCT116	Human colorectal cancer	0.53	0.40	0.33	0.39	0.55	0.68
KP-4	Human pancreatic cancer	1.00	0.81	0.86	0.80	0.71	0.73
TOV-21G	Human ovarian cancer	0.95	0.76	0.53	0.55	0.70	0.76
ES-2	Human ovarian cancer	0.93	0.66	0.61	0.56	0.64	0.67

**Table 6 pharmaceutics-16-01220-t006:** Characterization parameters of dual-loaded liposomes used in this study.

Preparation	Salt Gradient	Molar Ratio ^(1)^	Particle Size (nm) ^(2)^	PDI ^(2)^	ZP (mV) ^(2)^	EE% (Cyt) ^(3)^	EE% (Ida) ^(4)^
Lipo-1	Glu-Cu^2+^ (pH 7.4)	10:1	101.76 ± 1.57	0.063 ± 0.015	−35.28 ± 0.51	98.98 ± 0.85	95.59 ± 0.91
Lipo-2	Glu-Fe^2+^ (pH 7.4)	10:1	105.31 ± 2.01	0.073 ± 0.014	−35.19 ± 0.45	99.15 ± 0.71	95.25 ± 0.84
Lipo-3	(NH_4_)_2_SO_4_	10:1	103.61 ± 1.83	0.055 ± 0.016	−35.30 ± 0.57	99.63 ± 0.93	97.48 ± 0.81
Lipo-4	Glu-Cu^2+^ (pH 7.4)	20:1	102.19 ± 1.35	0.059 ± 0.012	−33.16 ± 0.50	98.81 ± 0.78	96.04 ± 0.75
Lipo-5	Glu-Cu^2+^ (pH 7.4)	30:1	101.28 ± 1.17	0.067 ± 0.010	−30.33 ± 0.40	98.92 ± 0.65	96.42 ± 0.88
Lipo-6	Glu-Cu^2+^ (pH 7.4)	40:1	104.35 ± 1.94	0.075 ± 0.013	−30.11 ± 0.42	99.17 ± 0.75	97.65 ± 0.61

(1) molar ratio = cytarabine/idarubicin; (2) after idarubicin loading; (3) EE%(Cyt): encapsulation efficiency of cytarabine; (4) EE%(Ida): encapsulation efficiency of idarubicin.

**Table 7 pharmaceutics-16-01220-t007:** In vitro stability of liposome formulation.

Day	Size (nm)	PDI	ZP (mV)	EE% (Cyt)	EE% (Ida)
0	101.28 ± 1.17	0.067 ± 0.010	−30.33 ± 0.40	98.92 ± 0.65	96.42 ± 0.88
7	102.90 ± 1.46	0.071 ± 0.013	−30.30 ± 0.42	98.53 ± 0.60	97.92 ± 0.75
14	103.08 ± 1.92	0.075 ± 0.018	−30.31 ± 0.40	98.35 ± 0.55	97.56 ± 0.77
28	103.81 ± 1.85	0.071 ± 0.022	−30.37 ± 0.41	98.89 ± 0.73	97.35 ± 0.80

**Table 8 pharmaceutics-16-01220-t008:** IC_50_ values of liposomes and combined free drugs in CCRF-CEM cells.

Compound	Absolute IC_50_ (mM)			Absolute IC_50_ (Mean ± SD, mM)
	Repeat 1	Repeat 2	Repeat 3	
Lipo-5 (30:1)	6686.69	4852.76	5941.06	5826.84 ± 922.29
Combined free drugs	232.16	222.50	193.72	216.12 ± 20.00

**Table 9 pharmaceutics-16-01220-t009:** PK parameters of CD-1 mice in plasma and bone marrow after intravenous injection of cytarabine-idarubicin liposomes or combined free drugs.

Matrix	Parameters	Cytarabine				Idarubicin			
		Lipo 1	Lipo 2	Lipo 3	Free Drugs	Lipo 1	Lipo 2	Lipo 3	Free Drugs
Plasma	T_max_ (h)	1.00	0.250	0.250	0.0830	1.00	1.00	0.250	0.0830
	C_max_ (ng/mL)	368,000	251,000	301,000	891,000	52,500	40,500	56,200	182
	AUC_0-t_ (h·ng/mL)	3,700,000	3,690,000	2,980,000	382,000	452,000	238,000	310,000	325
	AUC_0-∞_ (h·ng/mL)	3,700,000	3,700,000	2,990,000	382,000	452,000	238,000	312,000	400
	MRT (h)	7.26	11.0	8.63	0.364	5.14	3.98	4.14	2.49
	t_1/2_ (h)	4.24	6.92	6.19	5.82	2.19	3.41	3.40	3.61
	Vss (mL/kg)	23.6	36.2	35.4	593	30.0	44.0	36.3	20,300
	CL (mL/h/kg)	3.24	3.24	4.02	1570	5.82	11.0	8.42	4500
Bone marrow	T_max_ (h)	4.00	1.00	4.00	0.0830	4.00	4.00	4.00	4.00
	C_max_ (ng/mL)	17,900	18,800	16,500	517,000	8410	8810	9130	5450
	AUC_0-t_ (h·ng/mL)	323,000	468,000	287,000	336,000	120,000	140,000	167,000	35,700
	AUC_0-∞_ (h·ng/mL)	328,000	604,000	292,000	337,000	134,000	149,000	168,000	81,700
	MRT (h)	11.0	17.2	13.4	0.840	7.87	7.33	11.2	4.06
	t_1/2_ (h)	8.09	22.8	8.82	1.05	7.19	6.08	5.92	8.20

## Data Availability

The datasets used and/or analyzed during the current study are available from the corresponding author on reasonable request.

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
