# Peer review of "Enhanced Antitumor Efficacy of Cytarabine and Idarubicin in Acute Myeloid Leukemia Using Liposomal Formulation: In Vitro and In Vivo Studies"

_pharmaceutics, 2024, doi:10.3390/pharmaceutics16091220_

Round 1

Reviewer 1 Report

Comments and Suggestions for Authors

The research work by Zhu et al, “Dual drug liposomal formulation codelivery of cytarabine and idarubicin against acute myeloid leukemia” focuses on formulating a novel formulation to treat the acute myeloid leukemia through the use of dual-loaded liposomes containing cytarabine and idarubicin. Liposomal approached is well explored in recent era and many marketed formulation available. The researcher explored two molecules as codelivery. By optimizing the molar ratio of these drugs, the study demonstrates significant enhancements in antitumor efficiency. The liposomes show excellent stability, high drug entrapment efficiency, and rapid uptake by leukemia cells, indicating their potential to improve therapeutic outcomes. The findings suggest that this liposomal delivery system could offer a more effective treatment strategy for AML than traditional drug combinations. The comments are as follows:

1.       In the abstract and introduction, please discuss the need for this combination of cytarabine and  idarubicin drugs. A rationale should be clearly defined.

2.       Abstract, name of the method used to prepare liposome along with material (lipid) used.

3.       Abstract: The PK results showed the molar ratio of 23 cytarabine to idarubicin in plasma was maintained at 30:1 within 4 h. Please add the animal name used in the preclinical study; female mice

4.       Ref style should be [1], no superscript.

5.       Line 80-81. However, earlier data suggested increased leakage of idarubicin from cholesterol-80- 80 containing liposomes than from cholesterol-free[18,19,20]. Ref should be [18-20]

6.       Line 102-103: The dual-loaded liposomes of cytarabine and idarubicin were characterized in vitro and in vivo. Sentences need to be written with for in place of from. Please carefully proofread.

7.       Line: 91-92: DSPG, a negatively charged phosphatidylglycerol, was used to increase the circulation longevity of the liposomes. Is this also a aim of this work to make it long circulating?

8.       All methods, need to cite if it’s from literature.

9.       Table 1 Cells used in this experiment. Please cite this table content, based on what literature this cell selected?

10.   2.10 In vivo antitumor effect, this section need to write in better specially dose part, too much text unnecessary. Prepare table or figures for better understanding.

11.   If SEM/TEM data is available please share.

12.   Please ensure ethical permission for preclinical work is taken and shared with the publisher.

Reviewer 2 Report

Comments and Suggestions for Authors

The authors in the current research article (Dual drug liposomal formulation codelivery of cytarabine and idarubicin against acute myeloid leukemia) provide a comprehensive study about codelivery of cytarabine and idarubicin during treatment of acute myeloid leukemia. The present study is well-organized and comprehensively described. However, there are several issues that need to be addressed before acceptance for publication.

1.     Title: Please avoid word repetition during the selection of your title:

Dual drug = codelivery of cytarabine and idarubicin

Also, describe the actual work in the title

You perform in-vitro cytotoxicity and in-vivo study

Please use the following to improve the title:

ü  Enhanced Antitumor Efficacy of Cytarabine and Idarubicin in Acute Myeloid Leukemia using Liposomal formulation: In-vitro Cytotoxicity and In-vivo Studies

ü  Synergistic Antitumor Effects of Cytarabine and Idarubicin Delivered via Liposomal Formulation: In-vitro Cytotoxicity and In-vivo Evaluation.

The new title should be more descriptive and attract readers interested in the specific topic of the article.

2.     Please change (2.1 Lipid, drugs, and reagents) to (2.1.materials)

3.     In section 2.7.2 Determination of encapsulation efficiency (EE): please merge two equations in one equation and mention drug anonymous

4.      In Table 6, please add the standard deviation of mean IC50.

5.     In-vitro drug release is missed. If applicable, could you please perform an in-vitro release study to ensure that drugs are retained in liposomal formulation once exposed to a high volume of blood?

6.     Statistical analysis of data in the figure needs to be presented. This will make the figure readable without the need to read the manuscript.

7.     Discussion is missed.

Please try to justify the results obtained using the information presented in the literature. In addition, try to compare your results with the literature.

8.       The conclusion usually summarizes the paper's main outcomes and further work without mentioning the results. Please modify it.  

9.     The reference list needs to be updated. Please include at least 50% references from the last five years.

Reviewer 3 Report

Comments and Suggestions for Authors

The manuscript by Zhu et.al. describing “Dual drug liposomal formulation codelivery of cytarabine and idarubicin against acute myeloid leukemia” is a nicely written manuscript supported with adequate experimental data. This reviewer, however, found the Cryo-TEM experimental procedure and image shown is too good to believe, given the liposomal formulation with DSPC and DSPG lipids. Also Cryo-TEM cannot be done on a simple Cu grid, which is generally used for SEM imaging. 
more clarity would be required on this part to ascertain the genuineness of the work. 

Comments on the Quality of English Language

English language quality appears to be good

Reviewer 4 Report

Comments and Suggestions for Authors

In this submitted manuscript entitled “Dual drug liposomal formulation codelivery of cytarabine and idarubicin against acute myeloid leukemia”, the authors developed a series of liposomes co-encapsulating cytarabine and idarubicin to achieve synergistic treatment to leukemia. Generally, the study is well organized with proper experiment design and revealed nice outcomes. However, some improvements in the cell experiments are required before its publishment, as follows:

1.     Section 3.4, the cytotoxicity was evaluated for Lipo-5 while the uptake was evaluated in Lipo-1, please specify the reason to choose different formulations to be investigated in these experiments.

2.     Page 12, line 465, the authors claim the decrease of DiD and Hoechst staining were from the cell death. This explanation is not convincing. Especially for Hoechst staining, lower viability should indeed get higher signal intensity since the enhanced permeability of the dead cell. More proper interpretation of the result should be provided. Actually, I don’t think the Hoechst staining shows any significant variation among the different time points.

3.     Negative control of free drugs should be included in the cell uptake experiment to confirm the advantage of liposomes over free drugs.

Comments on the Quality of English Language

The language is generally readable but could be further polished by a native speaker.

Round 2

Reviewer 2 Report

Comments and Suggestions for Authors

Manuscript can be published now

Reviewer 3 Report

Comments and Suggestions for Authors

I have reviewed this manuscript a couple of weeks ago and the authors have done substantial improvements from the previous version. This reviewer has no objection to accept this ms for publication. 

Reviewer 4 Report

Comments and Suggestions for Authors

The authors addressed all my concerns. I don't have further comments.